# Measurement tools and indicators for assessing nurturing care for early childhood development: A scoping review

Joshua Jeong[1]*, Lilia Bliznashka[1], Eileen Sullivan[2], Elizabeth Hentschel[1], Youngkwang Jeon[2], Kathleen L. Strong[3], Bernadette Daelmans[3]

1 Department of Global Health and Population, Harvard T.H. Chan School of Public Health, Boston, MA, United States of America, 2 Harvard Graduate School of Education, Cambridge, MA, United States of America, 3 Department of Maternal, Newborn, Child and Adolescent Health and Ageing, World Health Organization, Geneva, Switzerland

* Joshua.jeong@hsph.harvard.edu

**Data Availability Statement:** All relevant data are within the manuscript or the supplementary files.

## Abstract

Nurturing care encompasses five components that are crucial for supporting early childhood development: good health, adequate nutrition, opportunities for early learning, responsive caregiving, and safety and security. While there has been increasing attention in global public health towards designing and delivering programs, services, and policies to promote nurturing care, measurement has focused more on the components of health and nutrition, with less attention to early learning, responsive caregiving, and safety and security. We conducted a scoping review to identify articles that measured at least one nurturing care outcome in a sample of caregivers and/or children under-5 years of age in low- and middle-income countries (LMICs). We systematically searched five electronic bibliographic databases for peer-reviewed articles published from database inception until November 30, 2020. We first classified outcomes to their respective nurturing care component, and then applied an inductive approach to organize key constructs within each nurturing care component and the specific measures and indicators used across studies. We identified 239 total articles representing more than 50 LMICs for inclusion in the review. The majority of included studies reported a measure of nutrition (N = 166), early learning (N = 140), and health (N = 102), followed by responsive caregiving (N = 78) and lastly safety and security (N = 45). For each nurturing care component, we uncovered multiple constructs relevant to children under-5: nutrition (e.g., anthropometry, complementary feeding), early learning (e.g., stimulation practices, early childhood education), health (e.g., birth outcomes, morbidity), responsive caregiving (e.g., parental responsivity, parent-child interactions), and safety and security (e.g., discipline, inadequate supervision). Particularly for outcomes of early learning and responsive caregiving, there was greater variability with regards to the measures used, reported indicators, and analytic construction of variables than the other three nurturing care components. This study provides a comprehensive review of the current state of measurement of nurturing care. Additional research is needed in order to establish the most optimal measures and indicators for assessing nurturing care, especially for early learning and responsive caregiving.

**Funding:** This work was funded by the World Health Organization (grant # 2020/1047163 awarded to J.J.). J.J. is supported, in part, by a Pathway to Independence Award from the Eunice Kennedy Shriver National Institute of Child Health and Human Development (K99HD105984). The funders had no role in study design, data extraction and analysis, decision to publish, or preparation of the manuscript.

**Competing interests:** The authors have declared that no competing interests exist.

## Introduction

Early childhood development (ECD) is defined as children's cognitive, physical, language, motor, and social and emotional development broadly spanning from birth to age 8 [1]. It forms the basis for health, learning, and wellbeing of a person throughout the lifecourse [2]. Globally, approximately two in five children under-5 years of age are at risk of not reaching their developmental potential [3, 4]. Over the past decades, there has been an acceleration in our understanding of the science that underpins young children's development, leading to greater knowledge about risk and protective factors as well as range of effective strategies for improving ECD and reducing inequities globally [5, 6]. This evidence has galvanized governments and a wide range of stakeholders to prioritize and invest in national and global programs and policies to promote ECD [7].

The United Nations Sustainable Development Goals (i.e., target 4.2 which calls for universal access to quality early childhood development, care, and pre-primary education) and the *Global Strategy for Women's, Children's and Adolescents' Health 2016–203*0 recognize ECD as a critical outcome for health and well-being throughout the life course. These global frameworks provide strategic directions for ensuring children not only survive, but also thrive so that they are able to transform societies to improve health and reach human potential [8]. In 2018, the World Health Organization (WHO), UNICEF and the World Bank, in collaboration with partners, launched the Nurturing Care Framework (NCF) as a roadmap for action, focusing especially on the critical period from conception until age 3 when key foundations are laid for children's future health and development [1].

To unlock their full potential, children need to receive *nurturing care*, meaning that they are raised in a stable caregiving environment that enables good health, adequate nutrition, opportunities for early learning, responsive caregiving, and safety and security [1]. The NCF describes essential policies and interventions, presents a universal progressive model of care, and proposes strategic action areas that are crucial for creating an enabling environment for families and caregivers to support young children's development. In particular, one strategic action of the NCF is to monitor progress, with the global milestone of "harmonized global indicators and measurement framework for nurturing care [that] are available and used to assess implementation and impact" [1]. Progress in this area requires a comprehensive mapping of measurement tools and indicators with respect to each component of nurturing care that have been used across LMICs.

To date, efforts towards harmonizing global indicators for young children have primarily focused on those related to health and nutrition. For example, there have been reviews on maternal and newborn health indicators [9, 10] and indicators for assessing infant and young child feeding practices [11]. Presently WHO and UNICEF have been coordinating global collaborations to further review, harmonize existing indicators, and identify gaps in indicators that require attention in the areas of maternal, newborn, child and adolescent health and nutrition [12].

On the other hand, measurement and indicators of early learning, responsive caregiving, and aspects of security and safety have not received sufficient attention in prior research. In recent years building upon the momentum of the Sustainable Development Goals and the Nurturing Care Framework, new efforts such as the ECD working group of Countdown to 2030 have advanced a core set of indicators relating to all nurturing care components in low- and middle-income countries (LMICs) [13]. The primary sources of the data are population-based surveys, namely the Multiple Indicator Cluster Surveys (MICS) and Demographic and Health Surveys (DHS). These efforts have shown the need for more standardized monitoring of nurturing care and in particular the lack of a population-level indicator for responsive

caregiving. However, measurement ambiguities for assessing nurturing care are not only of concern for population-level monitoring, but even more so in the context of program implementation and evaluation, especially for assessing early learning, responsive caregiving, and safety and security. In the absence of evidence-based guidance, a diverse and wide-ranging set of measures, indicators, and scoring methods are likely being used inconsistently across contexts and time, hampering programing monitoring and evaluation and national and global accountability and action.

To begin to address these gaps, we conducted a scoping review to summarize the measurement tools and indicators that have been used in the existing evidence to quantitatively operationalize the five components of nurturing care among children under-5 years of age in LMICs. Based on our review, we highlight trends and gaps in measurement and propose actions to inform future research, monitoring, implementation, and accountability for assessing programing for nurturing care globally.

## Methods

We conducted a scoping review, rather than a systematic review, because this evidence synthesis methodology is more appropriate for the nature of the present study that spans a heterogeneous literature regarding a concept that is very broad in scope [14]. We present findings in accordance with the Preferred Reporting Items for Systematic reviews and Meta-Analyses extension for Scoping Reviews (PRISMA-ScR) guidelines (see **S1 Checklist**) [15]. This scoping review was not preregistered.

### Search strategy

We searched electronic bibliographic databases (MEDLINE, Embase, CINAHL, Web of Science and Global Health Library) for peer-reviewed, published articles from database inception until November 30, 2020. A string of search terms combined keywords for concepts relating to child development, the five nurturing care components (i.e., health, nutrition, responsive caregiving, security and safety, and early learning), early childhood, and LMICs. The search string used in MEDLINE can be found as an example in **S1 Text**. These terms were modified and adapted for use in the other databases. Reference lists of included studies were scanned for any additional relevant studies that may have been missed.

### Study selection

Full-text articles were included if they met all the following criteria: (1) reported a quantitative indicator for any nurturing care component (i.e., health, nutrition, early learning, responsive caregiving, and security and safety), (2) targeted caregivers of children who were on average younger than aged five years, and (3) conducted in a LMIC. Studies were excluded if they targeted caregivers or children with diagnosed neurodevelopmental disorders or disabilities (e.g., autism) or were not empirical articles that reported metrics, measures, or indicators based on primary data collection (e.g., qualitative studies, protocol papers, systematic reviews). From this list of all eligible studies, we prioritized those that measured a responsive caregiving, early learning, or safety and security indicator (no restriction applied to year of publication) and extracted all relevant nurturing care indicators from those studies, including any nutrition and health indicators. For the remaining set of eligible studies that only measured a nutrition and health indicator in the context of ECD, we extracted all studies published since 2019 and a randomly-selected 10% subsample of remaining studies published before 2019, as a way of managing the large number of studies identified in the electronic databases reporting nutrition and health indicators.

## Data extraction

Four reviewers (JJ, ES, LB, YJ) were involved in the screening process of study titles and abstracts identified in the systematic search. Each study was independently screened by two reviewers using the web-based platform Covidence. Full texts of selected studies were reviewed to assess eligibility. Any discrepancy between the reviewers was resolved through discussion and consensus. Reference lists of included studies were examined to identify any potentially relevant publications not found through the electronic search.

Four reviewers (JJ, ES, LB, EH) independently extracted data from each eligible study using a structured extraction form in Excel. The main categories of data extracted for each study included: study design, sample, component of nurturing care measured, measurement tool, scoring and variable construction approach, and quality of the measure. JJ trained the reviewers over the course of a four-week training period (between September and October 2020) on how to use the data extraction sheet through a series of pilot exercises of pre-identified eligible articles, which each reviewer independently extracted. Any discrepancies were resolved through discussion and consensus; iterations were made to the extraction sheet as needed; and piloting continued until there were no changes needed to the extraction sheet and there was agreement in extractions across independent reviewers. In total, 20 articles were independently extracted by at least two reviewers and finalized during this training period. Thereafter, only one reviewer independently extracted each article. Weekly team meetings were held throughout the data extraction process (October 2020 to January 2021) to address any potential questions, which were resolved through discussion and consensus, and monitor data extraction progress.

## Data synthesis

We summarized the included studies by general study-level meta-data, including geography represented, year of publication, study purpose/objective (e.g., population monitoring vs individual assessment), and sample characteristics. For each reported outcome relevant to nurturing care, we first broadly determined which component it most related to (e.g., early learning, responsive care, safety and security) based on definitions from the NCF [1]. Then within each component, we identified any standardized measures. We define measures as a survey, scale, tool, or set of items that is designed to assess a particular concept. For example, the HOME Inventory is an example of a measure for parenting and the general quality of the caregiving environment [16]. After identifying the measures, we specified the indicator or variables that were constructed to quantify or operationalize the measure. For example, using the HOME Inventory, potential indicators could be the overall total score (across all subscales) or a particular subscale score (e.g., the parental responsivity subscale score). Finally, for further organization, we inductively grouped measures and indicators that represented similar constructs within a given nurturing care component. For example, if there were multiple measures that focused on whether children had play materials or books in the home, we could create a construct for "learning materials in the home" within the nurturing care component for early learning. We iteratively refined the list of constructs based on discussions among the research team. In this review, we present a narrative synthesis of results that summarize the various thematic constructs identified for each nurturing care component, the measures used, and the specific indicators and scoring approaches applied to the measures across studies. We did not assess quality or risk of bias for the included articles as the objective of this review was to more general in scope and aimed to describe the breadth of measurement tools and indicators that has been used in the literature.

### Ethics statement

All analyses were based on previously published studies. Therefore, no ethical approval or patient consent was required.

## Results

A total of 3,091 articles were identified from the electronic database search. An additional 8 were identified through other sources. A total of 239 articles met the eligibility criteria and were included in the scoping review (Fig 1). Characteristics for each study included are presented in S1 Table. Overall, the majority of articles that reported an outcome of nurturing care were in the context of program evaluations (54%), assessed caregivers and/or children during the first year of life (62%), and were mostly in Africa (30%), the Americas (19%), and the South East Asia (19%) regions (Table 1).

Out of the 239 articles extracted, nurturing care outcomes were most commonly represented for early learning (N = 140, 59%), then responsive caregiving (N = 78, 33%), and finally safety and security (N = 45, 19%). While different criteria were used for extracting nutrition and health outcomes thus limiting direct comparability, more than two-in-three articles measured a nutrition outcome (N = 166, 69%) and two-in-five measured a health outcome

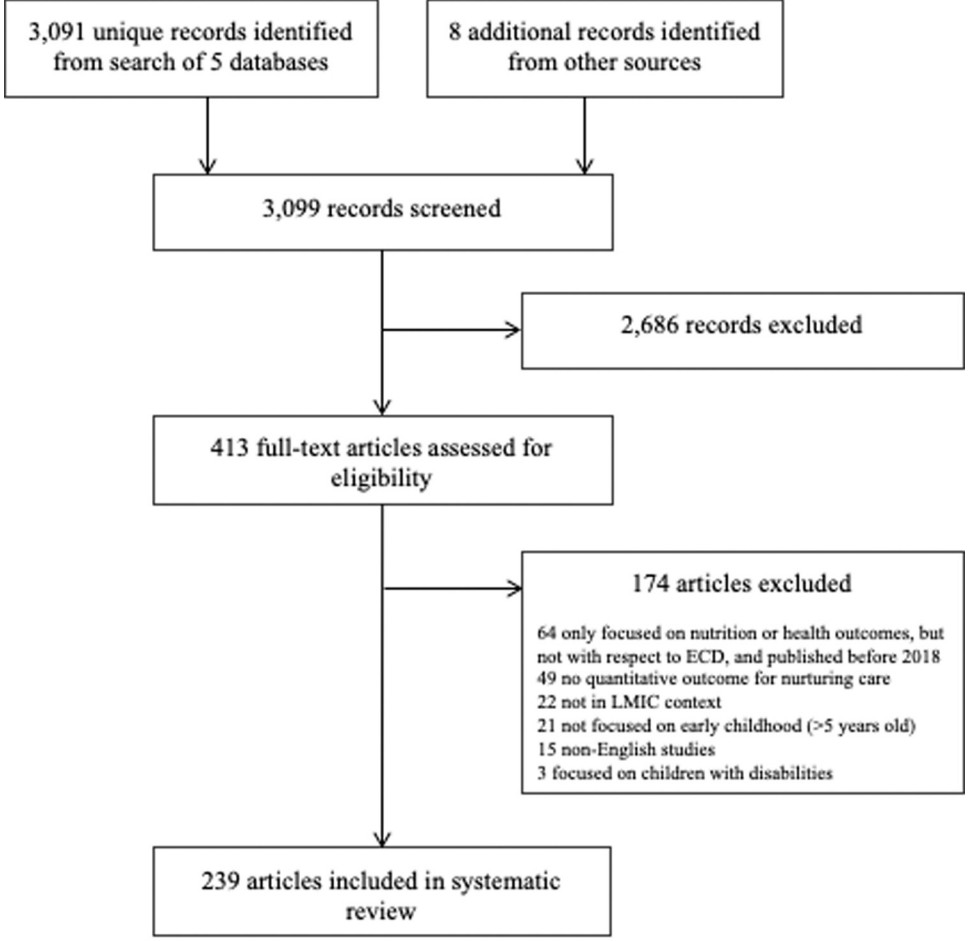

**Fig 1. PRISMA flow diagram of search results and process.**

**Table 1. Overview of study characteristics, by components of nurturing care indicators reported.**

| Study characteristics | All studies (N = 239) | | Early Learning (N = 110) | | Responsive Caregiving (N = 47) | | Early Learning/Responsive Caregiving (N = 43) | | Safety and Security (N = 45) | | Nutrition (N = 166) | | Health (N = 102) | |
|---|---|---|---|---|---|---|---|---|---|---|---|---|---|---|
| | % | N | % | N | % | N | % | N | % | N | % | N | % | N |
| **Original study design** | | | | | | | | | | | | | | |
| Population-based household survey (nationally representative) | 11% | 26 | 16% | 18 | 4% | 2 | 0% | 0 | 27% | 12 | 7% | 11 | 4% | 4 |
| Other household surveys (not nationally representative) | 26% | 62 | 27% | 30 | 26% | 12 | 12% | 5 | 36% | 16 | 19% | 32 | 27% | 27 |
| Program evaluation | 54% | 130 | 48% | 53 | 60% | 28 | 77% | 33 | 29% | 13 | 65% | 107 | 64% | 65 |
| Measurement development/validation study | 5% | 13 | 7% | 8 | 9% | 4 | 7% | 3 | 4% | 2 | 5% | 8 | 3% | 3 |
| Other[a] | 3% | 8 | 1% | 1 | 2% | 1 | 5% | 2 | 4% | 2 | 5% | 8 | 3% | 3 |
| **Youngest age at enrollment** | | | | | | | | | | | | | | |
| Prenatal | 9% | 21 | 2% | 2 | 17% | 8 | 14% | 6 | 9% | 4 | 9% | 15 | 10% | 10 |
| 0–1 year old | 62% | 147 | 61% | 67 | 53% | 25 | 61% | 26 | 47% | 21 | 67% | 111 | 73% | 74 |
| 1–3 years old | 16% | 39 | 18% | 20 | 28% | 13 | 19% | 8 | 18% | 8 | 17% | 28 | 15% | 15 |
| 3–5 years old | 13% | 32 | 19% | 21 | 2% | 1 | 7% | 3 | 27% | 12 | 7% | 12 | 3% | 3 |
| **Sample size:** | | | | | | | | | | | | | | |
| <100 | 10% | 23 | 6% | 6 | 21% | 10 | 7% | 3 | 7% | 3 | 6% | 10 | 3% | 3 |
| 101–500 | 34% | 82 | 28% | 31 | 40% | 19 | 47% | 20 | 24% | 11 | 36% | 59 | 40% | 41 |
| 501–1000 | 18% | 43 | 13% | 14 | 13% | 6 | 21% | 9 | 11% | 5 | 22% | 36 | 24% | 24 |
| 1001–2000 | 15% | 35 | 18% | 20 | 21% | 10 | 19% | 8 | 24% | 11 | 14% | 23 | 15% | 15 |
| >2000 | 23% | 56 | 36% | 39 | 4% | 2 | 7% | 3 | 33% | 15 | 23% | 38 | 19% | 19 |
| **Region of study population[b]** | | | | | | | | | | | | | | |
| Africa | 30% | 71 | 19% | 21 | 28% | 13 | 33% | 14 | 31% | 14 | 31% | 52 | 37% | 38 |
| Americas | 19% | 46 | 22% | 24 | 15% | 7 | 23% | 10 | 13% | 6 | 21% | 35 | 18% | 18 |
| South East Asia | 19% | 45 | 23% | 25 | 21% | 10 | 26% | 11 | 7% | 3 | 25% | 41 | 23% | 23 |
| Europe | 2% | 4 | 2% | 2 | 2% | 1 | 2% | 1 | 7% | 3 | 0% | 0 | 1% | 1 |
| Eastern Mediterranean | 7% | 16 | 6% | 7 | 21% | 10 | 12% | 5 | 9% | 4 | 6% | 10 | 7% | 7 |
| Western Pacific | 14% | 34 | 14% | 15 | 9% | 4 | 5% | 2 | 13% | 6 | 11% | 18 | 11% | 11 |
| Multiple regions | 10% | 23 | 15% | 16 | 4% | 2 | 0% | 0 | 20% | 9 | 6% | 10 | 4% | 4 |
| **Number of NCF components measured per study[c]** | | | | | | | | | | | | | | |
| 1 | 28% | 66 | 17% | 19 | 21% | 10 | - | - | 18% | 8 | 14% | 23 | 6% | 6 |
| 2 | 37% | 88 | 34% | 37 | 26% | 12 | - | - | 24% | 11 | 39% | 64 | 41% | 42 |
| 3 | 23% | 54 | 34% | 37 | 23% | 11 | - | - | 29% | 13 | 29% | 48 | 28% | 28 |
| 4 | 12% | 28 | 15% | 16 | 23% | 11 | - | - | 22% | 10 | 17% | 28 | 23% | 23 |
| 5 | 1% | 3 | 1% | 1 | 6% | 3 | - | - | 7% | 3 | 2% | 3 | 3% | 3 |

[a] Other study designs include for example routine health information systems and preschool-based surveys.

[b] Based on WHO classification of regions.

[c] This variable ranged from 0 to 5, with 5 representing a study that assessed all the NCF components. Any outcome that was originally included in the cross-cutting category of early learning/responsive caregiving was classified as assessing both early learning and responsive caregiving (2 components).

(N = 102, 43%). Of the five components of nurturing care, the median number of components assessed per study was two out of five, with the most commonly co-measured components being either nutrition and health (N = 89), or early learning and responsive caregiving (N = 58). For each nurturing care component, there was a consistent increase over the past decade in the number of published articles, with a striking growth particularly for early learning, responsive caregiving, and safety and security after 2016 (**Fig 2**).

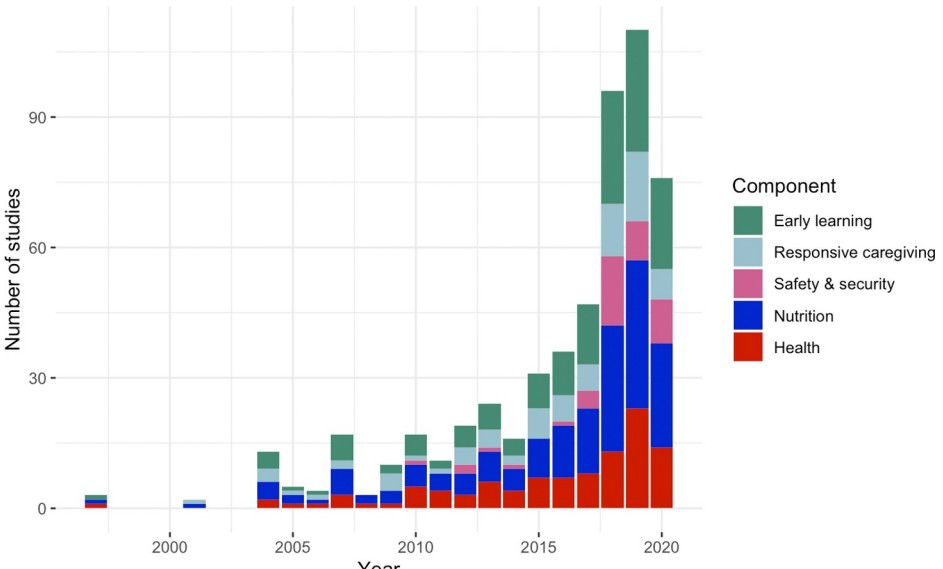

**Fig 2. Measurement of nurturing care outcomes among included studies over time, by the five components.** Note, the search was conducted in November, 2020, so trends for 2020 in the stacked bar graph may be incomplete.

**Table 2** summarizes the main constructs, measures, and indicators identified for each nurturing care component, which we elaborate upon further in the sections below. **Figs 3 and 4** are treemaps, which illustrate the hierarchical structure of the results organized broadly in terms of the nurturing care components. **Fig 3** shows the results for three of the nurturing care components–early learning, responsive caregiving, and safety and security–plus a fourth identified category that combined early learning and responsive caregiving. **Fig 4** shows the results for the remaining two nurturing care components of nutrition and health. We separated these into two figures because the results for nutrition and health are not directly comparable to the other nurturing care components, considering the different methodology applied for reviewing studies with respect to those two components.

## Early learning

One hundred ten studies measured the NCF component of *early learning* (**Table 3**). We identified four constructs: stimulation practices, learning materials, household stimulation, and early childhood education (ECE). Stimulation practices referred to activities that a caregiver engaged in with the child to promote early learning and development (e.g., reading, playing, naming things to child). Five different standardized measures were referenced, of which the Family Care Indicators (FCI) or an abbreviated version as used in the MICS was the most common. With this measure alone, we found seven different analytical approaches or indicators. A continuous variable for total number of stimulation activities was the most common (N = 14).

Learning materials referred to the presence or availability of learning materials for a child in the household, such as books and toys. We identified three standardized measures, and again the FCI or the abbreviated version as used in MICS were most often used. Specifically, an index score of the total number of play materials or books was the most common indicator (N = 8)

The third construct, household stimulation, represented a singular measure that intertwined both caregivers' engagement in stimulating activities (e.g., reading, playing) and

**Table 2. Summary table.**

| NCF component | Broad constructs | Main analytical approaches for monitoring and evaluation | Whether any data are publicly available at large- scale in LMICs | General measurement issues |
|---|---|---|---|---|
| Early learning | Stimulation practices | Index score, proportion, continuous (frequency), categorical, other (e.g., normalized, factor analysis, principal component analysis) | DHS, MICS | Measurement approaches (i.e., indicators) are highly variable across studies, even among those using the most common measure (FCI and HOME) |
| | Learning materials | Index score, proportion, categorical | DHS, MICS | Measurement approaches (i.e., indicators) are highly variable across studies, even among those using the most common measure (FCI and HOME) |
| | Household stimulation | Index score, continuous (weighted average) | None | Unclear whether this aggregated construct is conceptually valid or an improvement over reporting stimulation practices and learning materials separately |
| | Early childhood education | Proportion, continuous (preschool quality, duration of attendance), categorical | DHS, MICS | Most common was a single crude item for whether children attend early childhood education program (yes/no response); few measures about quality |
| Responsive caregiving | Parental responsivity | Continuous (sum, average) | None | Most commonly were direct observational tools; however the degree to which the tool specifically assessed parental responsivity varied considerably. Only one measure was identified as focusing primarily on responsivity (RIFL-P) |
| | Parent-child relationship | Continuous (sum, average) | None | Measures represented a mix of standardized scales (caregiver-reported), observational tools, and a few unstandardized/select item measures; these measures either did not directly assess parental responsivity or the exact parenting dimension evaluated was not entirely clear from the measure's description. Definition of what specific parenting behaviors constitute responsive caregiving is largely unclear in the literature |
| | Responsive feeding | Continuous (sum, average), proportion, categorical | None | The majority of measures were developed for a specific study and were not validated; measures varied in the degree to which they assessed parental responsivity during child feeding interaction versus context |
| Early learning & responsive caregiving | General caregiving environment | Continuous or proportion | None | Because the HOME inventory is a multidimensional measure (covering both early learning and responsive caregiving), studies that reported a HOME total score, could not be classified exclusively as either an early learning or responsive caregiving outcome |
| Safety and security | Disciplinary practices | Proportion, index | DHS, MICS | No major issues identified |
| | Maternal exposure to intimate partner violence | Proportion, index | DHS | No major issues identified |
| | Inadequate supervision | Proportion | MICS | Based on a single item included in MICS; yet validity is largely unknown |
| | Safe physical home environment | Continuous, proportion | None | Only one measure was identified (HOME) |
| | Birth registration | Proportion | DHS, MICS | No major issues identified |

(*Continued*)

**Table 2.** (Continued)

| NCF component | Broad constructs | Main analytical approaches for monitoring and evaluation | Whether any data are publicly available at large- scale in LMICs | General measurement issues |
|---|---|---|---|---|
| Nutrition | Anthropometry | Continuous, proportion | DHS, MICS | No major issues identified |
| | Complementary feeding practices | Index, proportion | DHS, MICS | Although indicators were originally developed for use with a defined age range of young children, studies use more broadly with older aged children. Thus, the validity with older aged children is unclear |
| | Breastfeeding practices | Proportion | DHS, MICS | Although indicators were originally developed for use with a defined age range of young children, studies use more broadly with older aged children. Thus, the validity with older aged children is unclear |
| | Food security | Index, proportion, categorical | None | No major issues identified |
| | Micronutrient status | Continuous, proportion | DHS, MICS | No major issues identified |
| Health | Birth outcomes | Continuous, proportion | DHS, MICS | No major issues identified |
| | Morbidity | Proportion | DHS, MICS | No major issues identified |
| | Hygiene and health practices | Index | DHS, MICS | Specific hygiene and health prevention practices varied across studies |
| | Healthcare utilization | Proportion | DHS, MICS | No major issues identified |
| | Mortality | Proportion | DHS, MICS | No major issues identified |

learning materials available to children in the household (e.g., books, toys), and did not report these two components separately as above. The FCI measure, and specifically a total score as the indicator, was the most common example of this case (N = 15). Finally, the last construct pertained to early childhood education (ECE), which was most frequently measured as a single item for whether children attended an ECE program (N = 26).

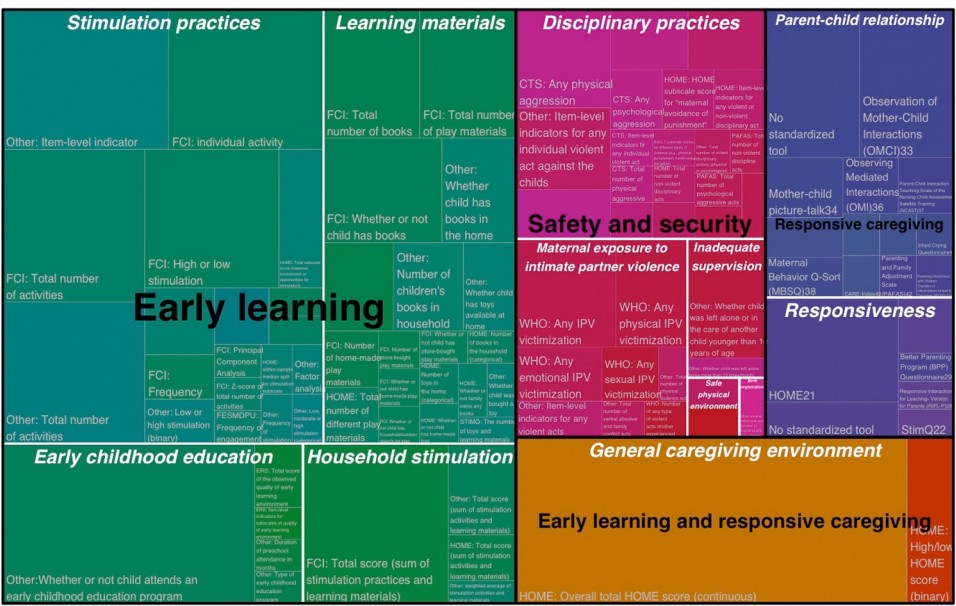

**Fig 3. Treemap of constructs and indicators hierarchically organized for early learning, responsive caregiving, and safety and security components of nurturing care.** Each rectangle represents a unique indicator that is nested in terms of three levels: nurturing care component (e.g., early learning), construct (e.g., learning materials), and indicator (e.g., numbers of books in the home). The size, location, and color of the rectangle is proportional to number of unique studies and the hierarchical structure.

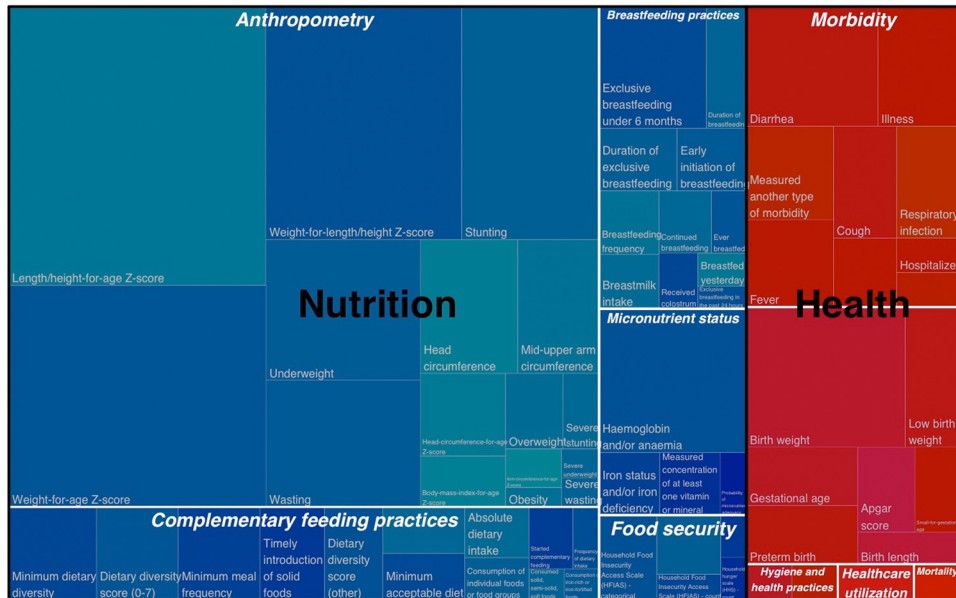

**Fig 4. Treemap of constructs and indicators hierarchically organized for nutrition and health components of nurturing care.** Each rectangle represents a unique indicator that is nested in terms of three levels: nurturing care component (e.g., nutrition), construct (e.g., anthropometry), and indicator (e.g., length/height-for-age z-score). The size, location, and color of the rectangle is proportional to number of unique studies and the hierarchical structure.

## Responsive caregiving

Forty-seven studies broadly measured *responsive caregiving* (**Table 4**). We uncovered significant variability in measurement tools for responsive caregiving and more specifically the degree to which these tools assessed responsive caregiving specifically versus other broader aspects of the parent-child relationship. To document this, we classified measures into three constructs: measures that specifically assessed parental responsivity to some degree, other measures that were more generally about the parent-child relationship but not technically responsiveness, and any measures that focused on responsive feeding in particular. For this component, we focused on summarizing the various measures used to assess responsive caregiving, but did not additionally document the specific analytical variables constructed from each measure, given that there was less variation in the indicators from a given measure.

Overall, the Responsive Interactions for Learning-Version for Parents (RIFL-P), was the only tool identified that primarily measured responsiveness (N = 1). Several other tools included subscales or few items assessing parental responsiveness, but these were part of a broader measure that assessed other dimensions of parenting besides responsiveness (e.g., HOME). The majority of identified tools assessed general caregiver-child relationships without a specific focus on responsiveness (N = 29), but they may have assessed another related parenting behavior such as sensitivity. For example, the Observation of Mother-Child Interactions tool was the most commonly used tool to assess parent-child relationships (N = 9). Finally, we identified a small number of studies that specifically measured responsive feeding (N = 7). The majority of responsive feeding measures were not validated and ranged considerably from direct observations of mother-child feeding interactions to brief survey asking mothers about how they encourage the child to eat when the child refuses.

**Table 3. Early learning.**

| Construct | Measure | Description of analytic variable (metric/indicator) | Overall scoring | Number of articles for a given scoring method | Child age range[a] |
|---|---|---|---|---|---|
| Stimulation practices | Family Care Indicators (FCI) [28] or abbreviated version as used in the Multiple Indicator Cluster Surveys (MICS) [25] | Total number of stimulating activities that caregiver reported taking part in with child (Usually the following six items: read books, told stories, sang, played, counted, and took child outside) | Index Score | 14 | 0–5 years |
| | | Item-level indicators of whether or not caregiver engaged with a stimulating activity with the child | Proportion | 14 | 0–7 years |
| | | High or low stimulation level (applying a cut-off score (e.g., engagement in 3 or 4 out of 6 total stimulating activities) | Proportion | 12 | 0–5 years |
| | | Frequency that caregiver reported engaging in play activities with child | Continuous | 2 | 0–2 years |
| | | Low, moderate or high stimulation as categorized from the total number of stimulating items that caregiver reported engaging in with child. | Categorical | 1 | 1–3 years |
| | | Principal Component Analysis of multiple item-level indicators of caregiver engagement in stimulating activities with the child | Continuous | 1 | 0–1 years |
| | | Z-score of total number of stimulating activities that caregiver reported engaging in with child | Continuous | 1 | 0–1 years |
| | Home Observation for Measurement of the Environment (HOME) or HOME-Short Form (HOME-SF) [16] | Total subscale score for either: (a) maternal involvement or (b) opportunities for stimulation subscales (without reporting a HOME total score, which we classify as a general measure of both early learning and responsive care) | Continuous | 4 | 1–3 years |
| | | Item-level indicators of whether or not caregiver reported or was observed engaging with a stimulating activity with the child | Proportion | 3 | 0–4 years |
| | | Cutoff score using a within-sample median split for either: (a) maternal involvement or (b) opportunities for stimulation subscales (without reporting a HOME total score, which we classify as a general measure of both early learning and responsive care) | Proportion | 1 | 0–5 years |
| | StimQ [29] | Total number of stimulating activities that the primary caregiver reported engaging in with child in the home | Index Score | 2 | 1–4 years |
| | Chinese Parent-Child Interaction Scale (CPCIS-8) [30] | Item level indicators of parent-child interactions and engagement in stimulating activities | Proportion | 1 | 4–5 years |
| | Family Environment checklist on Motor Development for Urban Preschool Children (FESMDPU) [31] | Frequency that caregiver reported engaging in activities related to teaching child and encouraging their development | Continuous | 1 | 3–5 years |
| | Developed for study (no reference to a standardized measure) | Item-level indicators of whether or not caregiver engaged in a certain stimulating activity with the child | Proportion | 15 | 0–2 years |
| | | Total number of activities caregiver reported engaging in with child | Index Score | 10 | 0–6 years |
| | | Low or high stimulation as classified by a cut-off score out of a total number of stimulating activities. | Proportion | 2 | 0–3 years |
| | | Low, moderate or high stimulation as categorized from the total number of stimulating items that caregiver reported engaging in with child | Categorical | 1 | 0–6 years |
| | | Frequency that caregiver reported engaging in stimulation activities with the child | Continuous | 1 | 0–6 years |
| | | Factor analysis of multiple item-level indicators about whether or not caregiver engaged in certain stimulation activities. | Continuous | 1 | 3–5 years |

(*Continued*)

**Table 3.** (*Continued*)

| Construct | Measure | Description of analytic variable (metric/indicator) | Overall scoring | Number of articles for a given scoring method | Child age range[a] |
|---|---|---|---|---|---|
| Learning materials | FCI [28] or abbreviated version as used in the MICS [25] | Total number of books (adult or children's/picture books) | Index Score | 8 | 0–7 year |
| | | Whether or not child has books (e.g., >0 or more than a given threshold like 3 or more books) | Proportion | 8 | 2–7 years |
| | | Total number of play materials (e.g., toys or objects meant for stacking; things for drawing; toys to play pretend games) | Index Score | 8 | 0–7 years |
| | | Adequate variety of play materials (Proportion of children having two or more types of playthings either homemade, store bought, or household objects used as toys) | Proportion | 4 | 3–5 years |
| | | Number of home-made play materials | Index Score | 2 | 0–7 years |
| | | Whether or not child has home-made play materials | Proportion | 1 | 3–4 years |
| | | Number of store-bought play materials | Index Score | 1 | 1.5 years |
| | | Whether or not child has store-bought play materials | Proportion | 1 | 3–4 years |
| | | Whether or not child has household objects (cups, bowls) or objects found outside that can be used for play | Proportion | 1 | 3–4 years |
| | HOME [16] | Total number of different play materials available for child at home. For example, the play materials subscale score of the HOME. | Index Score | 2 | 0–5 years |
| | | Categories of number of books in the household (none, 1–2, 3–5, > = 6) | Categorical | 1 | 1.5 years |
| | | Whether or not child has home-made toys | Proportion | 1 | 0–2 years |
| | | Number of toys in the home (none, 1–5 toys, 6–10 toys, 10+ toys) | Categorical | 1 | 0–3 years |
| | | Whether or not family owns any books | Proportion | 1 | 0–3 years |
| | StimQ [29] | The number of developmentally appropriate toys and learning materials that are available for child's use including symbolic play, art materials, fine motor/adaptive, language stimulating toys, and life size toys | Index Score | 1 | 1–3 years |
| | Developed for study (no reference to a standardized measure) | Whether child has books in the home | Proportion | 5 | 0–6 years |
| | | Number of children's books in household—presence of children's books or comic books in the household | Index Score | 4 | 0–3 years |
| | | Whether child was bought a toy in past 6 months | Proportion | 1 | 1–3 years |
| | | Whether child has toys available at home | Proportion | 3 | 0–6 years |
| Household stimulation | FCI [28] or abbreviated version as used in the MICS [25] | Total score including items for both parental engagement in stimulation activities and learning materials available in the home environment. This indicator aggregates stimulation practices and learning materials. | Index Score | 15 | 0–7 years |
| | HOME [16] | Total score including items for both parental engagement in stimulation activities and learning materials available in the home environment. This indicator aggregates the HOME subscales pertaining to early learning (e.g., maternal involvement, opportunities for stimulation subscales, materials) | Index Score | 2 | 0–3 years |
| | Developed for study (no reference to a standardized measure) | Total score of items relating to stimulation activities and learning materials available in the home | Index Score | 4 | 0–5 years |
| | | Caregiver investment score—weighted average of items relating to stimulation activities and learning materials in the home. | Continuous | 1 | 0–2 years |

(*Continued*)

**Table 3.** (Continued)

| Construct | Measure | Description of analytic variable (metric/indicator) | Overall scoring | Number of articles for a given scoring method | Child age range[a] |
|---|---|---|---|---|---|
| Early childhood education | Early Childhood Environment Rating Scales (ECERS) [32] or Infant/Toddler Environment Rating Scale (ITERS) [33] | Total mean score of the observed quality of early learning environment | Continuous | 2 | 0–5 years |
| | | Item-level indicators for subscales of the observed quality of early learning environment (e.g., structure, interactions, space) | Proportion | 1 | 0–5 years |
| | Developed for study (no reference to a standardized measure) | Whether or not child attends an early childhood education program | Proportion | 26 | 0–7 years |
| | | Type of early childhood education program (i.e., childcare centers, home-based community nurseries, other, none) | Categorical | 1 | 0–5 years |
| | | Duration of preschool attendance in months | Continuous | 1 | 3–5 years |

[a] Child age range refers to the sample assessed across the studies using a given metric, and not necessarily the age range of children for whom the tool was developed or could be used.

### Early learning/responsive caregiving

While the vast majority of indicators could be classified as pertaining to early learning or responsive caregiving, one notable exception was the HOME inventory, which is a multidimensional measure originally conceptualized according to six subdomains that broadly assess both early learning and responsive caregiving. Some articles reported each HOME subdomain score separately (often as a total subdomain score) which allowed us to classify the indicator to the respective non-overlapping component above (e.g., HOME responsivity subscale classified as an indicator for responsive caregiving). However, in most cases, articles reported an overall total HOME score spanning items across all six subdomains (pertain to both early learning and responsive caregiving), and therefore, the aggregated single indicator was considered as representing both early learning and responsive caregiving. Forty-three studies reported an overall HOME score as a continuous variable representing the sum total HOME score was the most common indicator (N = 41) and/or a proportion using some cutoff point to indicate high/low quality home environment (N = 5).

### Safety & security

Forty-five studies reported an indicator relevant to *safety and security* (Table 5). For this domain, we identified four main constructs: disciplinary practices, maternal exposure to intimate partner violence (IPV), inadequate supervision of the child, and birth registration. For disciplinary practices, we identified six standardized measures. The most common measure was the Parent-Child Conflict Tactics Scale as used in MICS, from which seven different types of indicators were reported across studies. The proportion of caregivers who used any physical punishment against the child was the most common indicator (N = 6).

Maternal IPV was most commonly measured using the Conflict and Tactics Scale or an adapted version as used in the WHO Multi-Country Study on Women's Health and Domestic Violence Questionnaire or the DHS. The most common indicator was the proportion of mothers who reported any form of IPV victimization (physical, emotional, and/or sexual violence) (N = 7).

Inadequate supervision of the child was measured using a one-item indicator that was predominantly collected as part of the MICS household survey (N = 5). Finally, birth registration

**Table 4. Responsive caregiving.**

| Construct | Measure | Description of measure[a] | Assessment Time for Observational Tools | Number of articles using a given measure | Child age range[b] |
|---|---|---|---|---|---|
| Parental responsivity | HOME [16] | Select subscales of the HOME include items about responsiveness. For example, the responsivity subscale includes: responding verbally to child's talk, permits child to play freely. The acceptance subscale includes: parent restricts or interferes with child's activity. Items include both caregiver reports and direct observation. As a measure of responsive care, studies that specifically reported HOME subscales scores for responsivity and acceptance are included (without a HOME total score, which we classify more broadly as a general measure of both early learning and responsive care). | Approximately 45–90 minutes for full HOME | 9 | 0–7 years |
| | Parent/Caregiver Involvement Scale (PCIS) [34] | This observational tool uses a free play interaction to assesses the affective status of the mother and child and the responsiveness of the mother to the infant's needs and initiations. It has 11 scales that assess physical and verbal interaction, responsiveness, play, teaching, control of activities, directives-demands, relationship, positive and negative statements, and goal setting. For each scale, the amount, quality, and appropriateness are determined and a score between 1 and 5 is assigned. Sensitivity is often calculated as the sum of physical involvement, verbal involvement, responsiveness, and positive verbal statements. | 10 minutes | 2 | 0–1 years |
| | Responsive Interactions for Learning- Version for Parents (RIFL-P) [35] | Unidimensional 11-item observational instrument designed to provide a rapid assessment of the extent to which a parent identifies and responds, incorporating sensitivity and stimulation, to the feelings and thoughts of the child with whom they are interacting. The RIFL-P measures three interconnected skills of the caregiver—(1) communicative clarity (providing meaningful verbal/nonverbal inputs to the child and fostering of shared understanding of the goals of the task); (2) mind-reading (thinking about what the child knows and understands); and (3) mutuality building (promoting reciprocity)—through a challenging task that elicits cooperation. Observers apply codes to each of the 11 items using a five-point Likert scale, ranging from 1 ("Not at all true") to 5 ("Very true"). A mean of the 11 items is calculated, yielding a composite score of responsivity that can range from 1 to 5. | Approximately 5–10 minutes | 1 | 1.5 years |
| | Better Parenting Program (BPP) Questionnaire [36] | The Better Parenting Program (BPP) questionnaire is a caregiver reported measure of caregiver knowledge and practices with regard to learning and holistic child development. Items were grouped together into sub-scales based on the content of the items. The conceptual analysis yielded three possible subscales, one of which was responsive parenting. | N/A | 1 | 2–7 years |
| | StimQ [29] | Caregiver reported measure, which includes a verbal responsivity subscale that measures verbal interactions between parents and the child and engagement in interactive play and talk while performing daily activities like bathing and feeding | N/A | 1 | 1–2 years |
| | Not a standardized tool [37] | This observational tool measures responsiveness, which is defined as responding contingently to the infant behaviors with talking, gazing, smiling. Behavioral composites are created by taking into account the length of the session and dividing the number of intervals the behavior was coded by the length of the observation time | Not specified | 1 | 0–1 years |
| | Not a standardized tool [38] | This observational tool measures maternal child-rearing practices and behaviors with 48 items. Maternal child rearing behaviors were scored and analyzed individually and classified under three behavioral categories, responsiveness, consistency and emotional stability. Items under responsiveness contained: mother breastfed child on demand, mother gives food whenever child demands it, mother soothes/strokes child while breastfeeding, mother soothes child to sleep if he awakens at night, and mother checks play-things for sharp edges before giving them to child for play | Not specified | 1 | 1–5 years |
| | Not a standardized tool [39] | Mothers are observed on mother–child dyadic typical routine and behavior. Maternal vocal responsiveness was calculated as the odds that a mother spoke to her infant within 2 seconds of the offset of the infant's non-distress vocalization relative to the odds that she spoke to her infant at other times. The contingency table for maternal verbal responsiveness tallied the 10ths of a second during which a mother: vocalized to the infant in the time windows after her infant vocalized non-distress, vocalized to the infant outside those time windows, did not vocalize to the infant in the time windows after her infant vocalized non-distress, and did not vocalize outside those time windows. | Not specified | 1 | 0–1 years |
| Parent-child relationship | Observation of Mother-Child Interactions (OMCI) [21] | The OMCI is an observational tool composed of 19 items that include: affect, touch, verbalization, sensitivity and contingent responses, scaffolding, language stimulation, focus, child affect, child focus, child's communication efforts, and mutual enjoyment. In particular, six items pertain to caregiver responsiveness: sensitivity and contingent responding (e.g., guiding the activity while also enabling independent exploration), scaffolding/expanding on the child's speech, pointing and naming objects in the book, posing questions to the child, responding to the child's questions or requests, and helping the child maintain interest. Sometimes positive affect and positive verbal statements are included in the responsiveness score as well. Caregivers are scored a 0 if a behavior was never/rarely observed, 1 if a behavior is sometimes observed, and 3 if a behavior is observed most of the time. Most commonly, an overall sum score is calculated across the 19 items. The 6 responsive items can also be selectively summed to create a total responsiveness score. | 5 minutes | 9 | 0–4 years |
| | Mother-child picture-talk [40] | Mothers are provided with a 2-sided page of pictures and instructed to talk about the pictures as they normally would. Observers assess if caregiver and child engage in directive talk (e.g., reading words or sentences, pointing out and naming objects), more advanced/responsive dialogic reading (e.g., asking children questions, expanding on their statements, asking them to elaborate or expand their statements, answering their questions, praising children) or were distracted or off-task. Frequencies for each utterance code were summed. | 5 minutes | 3 | 0–7 years |
| | In the Moment Coding [41] | Caregiver and child dyads are provided with a wooden puzzle, are asked to play together, and are observed. Maternal behaviors were coded in terms of positive regard, intrusiveness, disengagement, and sensitivity during the interactions. | 10 minutes | 2 | 2–7 years |
| | Observing Mediated Interactions (OMI) [42] | Caregiver child interactions are observed for 15 minutes and are scored for the total number of focusing (gaining the child's attention and directing them to the learning experience in an engaging manner), exciting (communicating emotional excitement, appreciation, and affection with the learning experience); expanding (making the child aware of how that learning experience transcends the present situation and can include past and future needs and issues); encouraging (emotional support of the child to foster a sense of security and competence); and regulating (helping direct and shape the child's behavior in constructive ways with a goal toward self-regulation) caregiver/child interactions. | 15 minutes | 2 | 2–5 years |
| | Parent-Child Interaction Teaching Scale of the Nursing Child Assessment Satellite Training (NCAST) [20] | Observational assessment whereby caregivers are guided to attempt a developmentally appropriate task that they have not yet seen their child perform (e.g., stack blocks), are provided with the materials for the task, and are asked to try and teach the task to their child, or mother and infant dyad might be observed during feeding. Four of the subscales measure the caregiver's contribution to the interaction (Sensitivity to Cues, Response to Distress, Social-Emotional Growth Fostering, and Cognitive Growth Fostering). Behaviors are scored "1" if they occur and "0" if they do not occur. Each interaction is scored on 76/73 items across six subscales. The number of scored behaviors observed is summed. Higher scores indicate a better quality of interaction. | Not specified | 2 | 3–4 years |
| | Maternal Behavior Q-Sort (MBSQ) [43] | Mothers and infants are given some sort of age-appropriate toy and observed. Trained researchers code maternal sensitivity using the 72-items describing potential maternal behaviors sorted into nine categories, with scores gradually ranging from "very unlike (1)" to "very similar (9)" to the observed mothers' behaviors, with each category containing eight items. The scores of these items represent the sensitivity level of maternal caregiving behavior in a family setting. Then, the observer's score is correlated with a standard score from an extremely sensitive mother, provided by the developers of the instrument. MBSQ scores thus range from −1 (least sensitive) to 1 (extremely sensitive) | 15–20 minutes | 2 | 0–3 years |
| | Adult/Child Interactive Reading Inventory (ACIRI) [44] | Interactive reading is assessed via observations of parent-child interactions while sharing a children's book. Coders observe parent-child dyads sharing a children's book and rate the dyad on 12 literacy behaviors related to the following: (1) enhancing attention to text, (2) promoting interactive reading and/or supporting comprehension, and (3) using literacy strategies. | 15–20 minutes | 1 | 2–4 years |
| | CARE-Index [45] | Caregivers are observed and rated in 7 areas including: facial expression, vocal expression, position and body contact, expression of affection, pacing of turns, control, and choice of activity. Mothers are given scores for each area that can range from 0 to 2. Sum scores to get a score for each domain (sensitivity, controlling, unresponsive). | 15–20 minutes | 1 | 0–1 years |

(*Continued*)

**Table 4.** (Continued)

| Construct | Measure | Description of measure[a] | Assessment Time for Observational Tools | Number of articles using a given measure | Child age range[b] |
|---|---|---|---|---|---|
| | Parenting Interactions with Children: Checklist of Observations Linked to Outcomes (PICCOLO) [19] | PICCOLO is a strengths-based observational checklist of 29 behaviors used to assess positive parenting interactions with children aged 10 to 47 months. PICCOLO items are clustered in four domains with seven to eight items per domain: affection (warmth, physical closeness, and positive expressions toward the child); responsiveness (respond- Mothers were given a task to do with their children. The mother sat on a yoga mating sensitively to a child's cues, needs, interests, and behaviors); encouragement (active and played with her child for five minutes. Mother is shown pictures of patterns on the support of play, exploration, curiosity, skills, and creativity); and teaching (shared conversations and play, cognitive stimulation, explanations, and questions). Items are coded on a three-point ordinal scale from "Absent", or not seen, to "Clearly" seen. | 10 minutes | 1 | 1.5 years |
| | Parenting and Family Adjustment Scale (PAFAS) [46] | PAFAS is a caregiver-reported measure of parenting, with higher scores on the PAFAS indicating higher levels of dysfunction, that is, higher consistency scores indicate less consistent parenting and higher coercive parenting scores indicate more coercive parenting. Domains include consistency, coercive parenting, encouragement, and parent-child relationship. | N/A | 1 | 2.5 years |
| | Infant Crying Questionnaire [47] | This caregiver-reported questionnaire measures negative emotions elicited by infant crying (i.e. worry, confusion, disappointment, helplessness, anger, guilt); interpretation of infant crying (i.e. hunger, not able to satisfy needs of infant, by nature my child cries easily, my baby would like to have company, I'm too tired to respond, something is hurting by baby) and active response to the infant crying (i.e. I talk to my baby with a calming voice and gestures, I do not do anything, I soothe my baby in my lap and with my arms, I call my spouse to help). Caregivers report 1 (does not fit at all) to 5 (fits exactly) | N/A | 1 | 0–1 years |
| | Father-Infant Interaction Scale [48] | Both fathers and mothers independently report the extent to which fathers engaged in 23 different activities, which represent three subscales: play (actively engaging in fun play and learning activities with the infant), caretaking (caring for the infant's physical needs), affection (demonstrating physical and emotional affection toward the infant, e.g., soothe and comfort baby, kiss baby). An index score is calculated which is the average of each Likert Scale | N/A | 1 | 0–1 years |
| | Not a standardized tool [49] | Caregiver- infant interactions are observed and videotaped during two-5-minute periods- book sharing and toy play. Caregivers are given a picture book and are asked to share it with their infant. Following this interaction, the book is removed and caregivers are given a shape-sorter toy and are asked to use it to play with their infant. Sensitivity is defined as awareness of the infant's direction of interest and their behavioral cues, and appropriate and timely responsiveness to them, and scored on a 1–5 Likert scale. All dimensions were event counts, apart from two that were rated low to high, on 1–5 scales (i.e., caregiver sensitivity and child attention). | 10 minutes total | 1 | 1–2 years |
| | Not a standardized tool [50] | * Individual caregiver-reported item for whether or not "caregiver responds to pleasing behavior by hugging and kissing child" | N/A | 1 | 0–5 years |
| | Not a standardized tool [51] | * Mothers reported which family members "help to soothe the baby when crying/upset" on a typical day (i.e., to generate a variable for whether or not the father helps to soothe the child when crying) | N/A | 1 | 0–1 years |
| Responsive feeding | Not a standardized tool [52] | Child and mother behaviors are observed during a midday meal when mothers are most likely to feed the child individually. Observer writes down all feeding behaviors of the mother and child, as well as foods eaten. Each meaningful unit of behavior in the transcripts received a code. Therefore, a frequency count of each behavior code is calculated for each mother child pair, reflecting the number of times the behavior occurred. Behaviors included: intended mouthfuls, mouthfuls, self-fed mouthfuls (Percentage of total mouthfuls that were self-fed), and child refusals. | Not specified | 2 | 0–2 years |
| | Not a standardized tool [53] | Mothers are observed and filmed for a 5-minute feeding session where they are asked to feed and interact with their infants as they would do at home. Recordings are rated on five dimensions of maternal sensitivity and emotional availability. A trained coder scores the five-minute video-recordings, using five-point scales. The five dimensions of sensitivity are: mothers' expression of love (i.e., mother is positively engaged, animated and vocalizes positively during interaction); eye contact between mother and infant; extent to which mothers followed their infants' cues (i.e., mother is attuned to infant's subtle behaviors and respond to them appropriately); communication between mother and infant (i.e., mother interprets infant's communications accurately and responds to them promptly and appropriately); synchronous interactions (i.e., joint orientation and coordination of actions between the infant and mother). | 5 minutes | 1 | 0–2 years |
| | Not a standardized tool [54] | * Quality of feeding interaction is measured using the observed feeding index. Very low-quality means that the following was observed during the feeding episode: < = 1 positive talk by mother towards child, and < = 1 episodes of playful feeding and < = 1 responsive feeding actions, plus one or more negative actions such as force feeding, holds child's head still to give food, shaking, threatening, shouting or berating observed by the mother towards child during feeding session. Responsive feeding action is not defined. | Not specified | 1 | 1–2 years |
| | Not a standardized tool [55] | Caregiver–child interactions while feeding are observed and video coded. Intended bites (the caregiver or child brought food to within a few inches of the child's mouth with the intention of it entering the mouth for consumption, whether or not it was consumed) are evaluated. All indicators are coded at level of intended bite. The person who is feeding the child, verbalization of the caregiver, and physical actions of the caregiver, are also measured | Not specified | 1 | 1–2 years |
| | Not a standardized tool [25] | * Mothers are asked: "What do you usually do to get your child to eat?" Response and scoring options were one of the following categories: (a) Nothing/tell child to eat, (b) encourage/praise, (c) force/threaten/hit, (d) nothing/good appetite, or (e) other | N/A | 1 | 0–5 years |
| | Not a standardized tool [56] | * Child-responsive feeding practices are measured by asking the mother what she usually does when the child refuses to eat. A mother who reports more than one behavior consistent with responsive feeding (for example, feeding patiently, talking with child, reducing distractions, changing what is fed, letting child self-feed, or encouraging child) and who did not force the child to eat were coded as one. Mothers who displayed inconsistent behavior with responsive feeding and/or forced the child to eat were coded as zero. | N/A | 1 | 0–4 years |

[a] The majority of studies used standardized scales comprising of multiple items to directly or indirectly assess responsive caregiving. There was more variation in the specific scales/measures used than the analytic variable reported based on a given scale (which was most commonly an overall sum score or average across the multiple items). Therefore, for responsive caregiving, we describe details about the measurement construct/procedure rather than the scoring of the analytic variable.

* The asterisk denotes a metric of responsive caregiving that was based on a dichotomous or categorical variable.

[b] Child age range refers to the sample assessed across the studies using a given metric, and not necessarily the age range of children for whom the tool was developed or could be used.

**Table 5. Safety and security.**

| Construct | Measure | Description of analytic variable (metric/ indicator) | Scoring/ reported variable | Number of articles for a given scoring method | Child age range[a] |
|---|---|---|---|---|---|
| Disciplinary practices | Parent-Child Conflict Tactics Scale [57] (as used in MICS) | Any physical aggression—Caregiver use of any physical punishment (e.g., spanking with bare hand, hitting with object) | Proportion | 6 | 0–6 years |
| | | Any non-violent discipline—Caregiver use of any actions to respond to a child's challenging behaviors without using violence (e.g., took away privileges, explained wrong behavior, gave something else to do) | Proportion | 3 | 0–4 years |
| | | Any violent discipline—Caregiver use of any form of violent discipline, which includes any physical punishment and/or psychological aggression | Proportion | 3 | 0–4 years |
| | | Any psychological aggression—Caregiver use of any psychological aggression (e.g., calling child offensive names, shouting/screaming at child) | Proportion | 2 | 2–6 years |
| | | Total number of physical aggressive acts caregiver used against a child | Index score | 1 | 4 years |
| | | Total number of violent disciplinary actions (physical or psychological) caregiver used against a child | Index score | 1 | 4–6 years |
| | | Item-level indicators of whether or not caregiver engaged in any individual violent act against the child | Proportion | 1 | 0–5 years |
| | HOME [16] | Total number of violent disciplinary actions (physical or psychological) caregiver used against a child (e.g., criticizing/ shouting, and threatening/ hitting/pushing/spanking) | Index score | 3 | 0–4 years |
| | | HOME subscale score for "maternal avoidance of punishment" | Continuous (average subscale score) | 2 | 0–5 years |
| | | Item-level indicators of whether or not caregiver engaged in any violent or non-violent disciplinary act against the child | Proportion | 2 | 0–6 years |
| | | Total number of non-violent disciplinary actions caregiver used against a child (e.g., Guide or give positive discipline, explain without being upset) | Index score | 1 | 0–3 years |
| | Parent and Family Adjustment Scales (PAFAS) [58] | Total number of non-violent discipline acts | Index score | 1 | 0–2 years |
| | | Total number of psychological aggressive acts | Index score | 1 | 0–2 years |
| | Physical Punishment Questionnaire (PPQ) [59] | Frequency of how often caregivers used physical aggression | Continuous/ Likert score | 1 | 4–6 years |
| | Disciplinary Style Questionnaire (DSQ) [60] | The DSQ is comprised of 7 subscales: inductive discipline, manipulating privileges, physical punishment, harsh verbal discipline, argument, shaming, and ignoring. | Index score | 1 | 2–7 years |
| | Socolar Discipline Survey [61] | Two questions regarding the frequency of spanking and of slapping the child's hand. | Index score | 1 | 2–4 years |
| | Developed for study (no reference to a standardized measure) | Item-level indicators of whether or not caregiver engaged in any individual violent act against the child | Proportion | 5 | 0–4 years |
| | | Any psychological aggression—Caregiver use of any psychological aggression (e.g., calling child offensive names, shouting/screaming at child) | Proportion | 1 | 3–5 years |

(*Continued*)

**Table 5.** (Continued)

| Construct | Measure | Description of analytic variable (metric/indicator) | Scoring/ reported variable | Number of articles for a given scoring method | Child age range[a] |
|---|---|---|---|---|---|
| Maternal exposure to intimate partner violence | WHO Multi-Country Study on Women's Health and Domestic Violence Questionnaire [62] (as used in DHS) | Any IPV victimization—Maternal report of any form of IPV victimization, which includes any physical, emotional, and/or sexual violence | Proportion | 7 | 0–5 years |
| | | Any physical IPV victimization—Maternal report of physical IPV victimization (e.g., beaten, punched) | Proportion | 5 | 0–6 years |
| | | Any emotional IPV victimization—Maternal report of emotional IPV victimization (e.g., humiliated in front of others, threatened) | Proportion | 3 | 0–5 years |
| | | Any sexual IPV victimization—Maternal report of sexual IPV victimization (e.g., forced you to have sexual intercourse against will) | Proportion | 2 | 0–5 years |
| | | Score for number of violent acts mother experienced by intimate partner (any physical, emotional, and/or sexual violence), with each item rated on 1–4 scale | Continuous | 1 | 3–4 years |
| | Developed for study (no reference to a standardized measure) | Total number of physical violence acts mother experienced by intimate partner | Index score | 1 | 1–5 years |
| | | Total number of verbal abusive and family conflict acts experienced by mother | Index score | 1 | 1–5 years |
| | | Item-level indicators of whether or not caregiver experienced any violent act against them by intimate partner | Proportion | 2 | 0–1 year |
| Inadequate supervision | Not a standardized measure (as used in the MICS) | Item for whether child was left alone or in the care of another child younger than 10 years of age | Proportion | 6 | 0–6 years |
| | HOME [16] | Item-level indicator for whether or not child was left alone home more than 10 times/month | Proportion | 1 | 1–4 years |
| Safe physical environment | HOME [16] | HOME subscale score for "organization of environment" | Continuous | 1 | 0–1 year |
| | | HOME subscale score for "organization of environment"–applying cutoff for above/below within-sample median | Proportion | 1 | 0–5 years |
| Birth registration | Not a standardized measure (as used in the MICS) | Item for whether child's birth was reported as registered with civil authorities | Proportion | 1 | 0–5 years |

[a] Child age range refers to the sample assessed across the studies using a given metric, and not necessarily the age range of children for whom the tool was developed or could be used.

was reported in 1 study using a single-item indicator collected as part of the MICS household survey.

## Nutrition

We identified 166 total studies which reported at least one indicator for *nutrition* (Table 6). We grouped indicators into five constructs: anthropometry, breastfeeding practices, complementary feeding practices, micronutrient status, and food security. For anthropometry, all indicators were standardized with 16 indicators based on the 2006 WHO Child Growth Standards and 4 indicators based on 1977 National Center for Health Statistics (NCHS) Growth Curves for Children. The most frequently reported anthropometry indicator was length/height-for-age Z-score in the majority of studies (N = 125).

**Table 6. Nutrition.**

| Nutrition domain | Measure | Description of analytic variable (metric/indicator) | Overall scoring | Number of articles for a given scoring method | Child age range[a] |
|---|---|---|---|---|---|
| Anthropometry | Length/height-for-age Z-score | Age- and sex-specific Z-score calculated based on weight and height directly assessed | Continuous | 125 | 0–96 months |
| | Weight-for-age Z-score | Age- and sex-specific Z-score calculated based on weight and height directly assessed | Continuous | 100 | 0–72 months |
| | Weight-for-length/height Z-score | Age- and sex-specific Z-score calculated based on weight and height directly assessed | Continuous | 80 | 0–84 months |
| | Stunting | Length/height-for-age Z-score < -2 SD | Binary | 59 | 0–83 months |
| | Underweight | Weight-for-age Z-score < -2 SD | Binary | 40 | 0–72 months |
| | Wasting | Weight-for-length/height Z-score < -2 SD | Binary | 36 | 0–72 months |
| | Head circumference | Continuous measure in cm, directly assessed | Continuous | 24 | 0–61 months |
| | Mid-upper arm circumference | Continuous measure in cm, directly assessed | Continuous | 20 | 6–60 months |
| | Head-circumference-for-age Z-score | Age- and sex-specific Z-score calculated based on directly assessed head circumference | Continuous | 13 | 0–24 months |
| | Body-mass-index-for-age Z-score | Age- and sex-specific Z-score calculated based on weight and height directly assessed | Continuous | 8 | 0–28 months |
| | Overweight | Weight-for-length/height Z-score >2 SD | Binary | 8 | 0–60 months |
| | Length/height-for-age Z-score | Age- and sex-specific Z-score calculated based on weight and height directly assessed | Continuous | 6 | 0–61 months |
| | Severe stunting | Length/height-for-age Z-score < -3 SD | Binary | 5 | 6–20 months |
| | Arm-circumference-for-age Z-score | Age- and sex-specific Z-score calculated based on directly assessed mid-upper arm circumference | Continuous | 4 | 6–60 months |
| | Weight-for-age Z-score | Age- and sex-specific Z-score calculated based on weight and height directly assessed | Continuous | 4 | 0–61 months |
| | Weight-for-length/height Z-score | Age- and sex-specific Z-score calculated based on weight and height directly assessed | Continuous | 4 | 0–61 months |
| | Severe underweight | Weight-for-age Z-score < -3 SD | Binary | 2 | 6–20 months |
| | Severe wasting | Weight-for-length/height Z-score < -3 SD | Binary | 2 | 6–20 months |
| | Obesity | Weight-for-length/height Z-score >3 SD | Binary | 2 | 0–60 months |
| Complementary feeding practices | Minimum dietary diversity | Proportion of children who consumed ≥4 food groups in the past 24 hours, based on caregiver report | Binary | 15 | 0–72 months |
| | Minimum meal frequency | Proportion of children who received solid, semi-solid or soft foods the minimum number of times or more in the past 24 hours. Some studies reported the number of meals in the previous day | Binary, Count | 14 | 6–24 months |
| | Dietary diversity score | Summary score of the number of food groups consumed by the child in the past 24 hours based on caregiver report, range 0–7 | Count | 14 | 0–36 months |
| | Timely introduction of solid foods | Proportion of children who started receiving solid, semi-solid, or soft foods at 6 months of age, based on caregiver report | Binary | 11 | 6–72 months |

(*Continued*)

**Table 6.** (Continued)

| Nutrition domain | Measure | Description of analytic variable (metric/indicator) | Overall scoring | Number of articles for a given scoring method | Child age range[a] |
|---|---|---|---|---|---|
| | Dietary diversity score | Summary scores of the number of food groups consumed by the child in the past 24 hours based on caregiver report, not applying the WHO IYCF food groups, ranges 0–8, 0–9, 0–12 | Count | 10 | 6–72 months |
| | Minimum acceptable diet | Proportion of children who received a minimally acceptable diet in the past 24 hours, i.e., meet minimal meal frequency and have a dietary diversity score ≥4. | Binary | 7 | 6–24 months |
| | Age of introduction of first foods | Age of introduction of first foods defined as categorical or count variable | Categorical, Count | 7 | 6–8 months |
| | Absolute dietary intake | Absolute intake of micronutrients and macronutrients, including from breast milk | Continuous | 6 | 3–72 months |
| | Consumption of individual foods or food groups | Intake in the past 24 hours of different food groups: dairy; meat/fish/eggs; meat; vitamin A-rich foods; non-meat protein (e.g. soy, eggs, beans), breakfast meal, roller meal | Binary | 5 | 0–24 months |
| | Started complementary feeding | | Binary | 5 | 6–8 months |
| | Frequency of dietary intake | Frequency (# of days/week) of intake of micronutrient-rich vegetable and animal sourced foods during the previous week | Count, Categorical | 3 | 0–17 months |
| | Consumption of iron-rich or iron-fortified foods | Proportion of children who received an iron-rich food or a food that was specially designed for infants and young children and was fortified with iron, or a food that was fortified in the home with a product that included iron during the previous day | Binary | 2 | 6–24 months |
| | Consumed solid, semi-solid, soft foods | Proportion of children who consumed solid, semi-solid, or soft foods in the previous day | Binary | 2 | 6–18 months |
| Breastfeeding practices | Exclusive breastfeeding under 6 months | Proportion of children <6 months of age who were fed exclusively with breast milk. Some studies calculated at individual time points or created new binary variables for specific duration (e.g. EBF ≥3 months vs. EBF <3 months) | Binary | 24 | 0–72 months |
| | Duration of exclusive breastfeeding | Different types of definitions for duration of exclusive breastfeeding, including number of months, median number of months | Count | 9 | 1–12 months |
| | Duration of breastfeeding | Different types of definitions for duration of breastfeeding, including categorical variables, number of months, median number of months | Categorical, Count | 8 | 0–24 months |
| | Early initiation of breastfeeding | Proportion of caregivers who initiated breastfeeding within 1 hour of delivery | Binary | 8 | 0–72 months |
| | Breastfeeding frequency | Different types of definitions/groupings of the categories and references periods. | Binary, Count | 7 | 3–18 months |
| | Breastmilk intake | Different types of definitions for intake of breastmilk or formula in the past 24 hours | Binary, Continuous | 6 | 6–18 months |
| | Continued breastfeeding | Continued breastfeeding at 1 year and 2 years per the WHO IYCF indicators or some other age range not per the WHO IYCF indicators | Binary | 6 | 36–72 months |
| | Received colostrum | Proportion of children who were given colostrum | Binary | 4 | 1–72 months |
| | Ever breastfed | Proportion of children who were ever breastfed | Binary | 4 | 0–59 months |
| | Breastfed yesterday | Proportion of children breastfed in the previous 24 hours | Binary | 3 | 6–20 months |

(*Continued*)

**Table 6.** (Continued)

| Nutrition domain | Measure | Description of analytic variable (metric/indicator) | Overall scoring | Number of articles for a given scoring method | Child age range[a] |
|---|---|---|---|---|---|
| | Exclusive breastfeeding in the past 24 hours | The proportion of children fed only breast milk in the past 24 hours, based on maternal recall | Binary | 2 | 0–24 months |
| Food security | Household hunger scale (HHS) | A HHS score (range 0–6) is calculated based on the responses to 3 questions with 4-likert type response options. | Count | 2 | 0–42 months |
| | | HHS score is broken down into categories to define little to no hunger (score 0–1), moderate hunger (score 2–3), severe hunger (score 4–6), which are then used to classify households as deprived/insecure (score >1) or food secure (score 0 or 1). | Binary | 2 | 6–18 months |
| | Household Food Insecurity Access Scale (HFIAS) | HFIA category is calculated based on the frequency-of-occurrence during the past four weeks for the 9 food insecurity-related conditions: 1 = Food Secure, 2 = Mildly Food Insecure Access, 3 = Moderately Food Insecure Access, 4 = Severely Food Insecure Access. | Categorical | 9 | 0–48 months |
| | | Binary variables created from the HFIA category are also reported for % of households experiencing certain type of food insecurity based on the categories | Binary | 7 | 0–48 months |
| | | A HFIAS score (range 0–27) is calculated as the sum of the frequency-of-occurrence during the past four weeks for the 9 food insecurity-related conditions. | Count | 3 | 0–48 months |
| Micronutrient status | Hemoglobin and/or anemia | Assessed hemoglobin directly from a finger or heel prick. Anemia defined as hemoglobin > 11 g/dL. | Continuous (hemoglobin), binary (anemia) | 39 | 3–72 months |
| | Iron status and/or iron deficiency | Assessed iron status or blood iron levels using serum transferrin receptor, serum/plasma ferritin, body iron status, free erythrocyte protoporphyrin, or mean corpuscular volume. Iron deficiency defined based on serum transferrin receptor or plasma ferritin levels | Continuous (iron status), binary (iron deficiency) | 7 | 0–59 months |
| | Measured concentration of at least one vitamin or mineral | Assessed blood or urine concentration of at least one vitamin or mineral from the following: iodine, vitamin A, vitamin E, vitamin B-12, selenium, zinc, folate | Continuous | 7 | 6–18 months |
| | Probability of micronutrient adequacy | Probability of adequacy is calculated as the probability that a child's usual intake is above EAR | Binary | 3 | 6–72 months |

[a] Child age range refers to the sample assessed across the studies using a given metric, and not necessarily the age range of children for whom the tool was developed or could be used.

Four of the 11 indicators for breastfeeding practices and 6 of the 13 indicators for complementary feeding practices were based on the 2008 WHO Infant and Young Child Feeding Indicators. The rest of the indicators for breastfeeding and complementary feeding practices were not standardized. While nearly all other indicators were used across studies among samples of children ranging broadly from 0–5 years of age, complementary feeding practices were particularly assessed in children 6–24 months of age.

We identified 4 indicators in the micronutrient status group, two of which were based on direct assessment of blood samples. Lastly, we identified 2 indicators of food security based on standardized measures: the Household Hunger Scale (HHS) and the Household Food Insecurity Access Scale (HFIAS). Both were reported as count scores, categorical variables, or binary variables, depending on the purposes of the studies.

**Table 7. Health.**

| Health domain | Measure | Description of analytic variable (metric/indicator) | Overall scoring | Number of articles for a given scoring method | Child age range[a] |
|---|---|---|---|---|---|
| Birth outcomes | Birth weight | Weight at birth reported by the caregiver or observed/recorded from the child's health card | Continuous | 41 | 0–61 months |
| | Low birth weight | A binary indicator for whether the child's weight at birth was <2500 grams | Binary | 13 | 0–61 months |
| | Gestational age | Gestational age of the child reported by the mother | Count | 12 | 0–72 months |
| | Preterm birth | A binary indicator for whether the child was born before 37 weeks of gestation, based on maternal report | Binary | 11 | 0–36 months |
| | Apgar score | A count score of certain perinatal vitals, usually assessed at 1 and 5 minutes after birth. Some studies defined categorical or binary variables breaking down the overall score. Lower score indicates need for extra or emergency care | Count, Categorical, Binary | 8 | 0–24 months |
| | Small-for-gestational age | Defined as a birth weight below the 10th percentile for gestational age based on the sex- specific curves | Continuous | 7 | 0–39 months |
| | Birth length | Length, reported | Continuous | 6 | 0 months |
| Morbidity | Diarrhea | Maternal/caregiver report of whether the child had diarrhea over a set recall period, usually 1 or 2 weeks but varies | Binary | 29 | 0–59 months |
| | Illness | Maternal/caregiver report of whether the child was sick/ill over a set recall period, usually 1 or 2 weeks but varies, or the number of days the child was sick | Binary, Count | 18 | 0–42 months |
| | Fever | Maternal/caregiver report of whether the child had fever over a set recall period, usually 1 or 2 weeks but varies | Binary | 14 | 0–42 months |
| | Measured another type of morbidity | Maternal/caregiver report of whether the child had another type of morbidity specific to the study outcomes (e.g., convulsions, seizures, dysentery, fetal alcohol spectrum disorder, vomiting) | Binary | 14 | 0–72 months |
| | Cough | Maternal/caregiver report of whether the child had cough over a set recall period, usually 1 or 2 weeks but varies | Binary | 13 | 0–59 months |
| | Respiratory infection | Maternal/caregiver report of whether the child had respiratory infection (lower, upper, or acute) over a set recall period, usually 1 or 2 weeks but varies | Binary | 13 | 0–48 months |
| | HIV status | Maternal/caregiver report or direct test of whether the child has HIV and/or measured viral load and/or CD4 count | Continuous (viral load, CD4 count), Binary (status) | 8 | 0–72 months |
| | Hospitalized | Maternal/caregiver report of whether the child was hospitalized or re-hospitalized for any reason over a set recall period, varying by study | Binary | 4 | 6–59 months |
| | Inflammation | Assessed at least one biomarker (e.g. C-reactive protein) and/or reported inflammation based on a standardized cut-off | Continuous (biomarker), binary (inflammation) | 4 | 6–60 months |
| Hygiene and health practices | Assessed at least one type of household water, hygiene, and sanitation practice | Maternal report or direct observation of household water, hygiene, and sanitation practices, including mother and child cleanliness | Count | 3 | 0–48 months |
| | Preventive health practices | Mother/caregiver reported on whether the household had access to the safe water, latrine use to dispose of children's feces, the child had received immunizations (BCG, DPT, polio, measles), and the child had received Vitamin A drops. Higher score indicated more preventive practices. | Count | 3 | 4–36 months |

*(Continued)*

**Table 7.** (Continued)

| Health domain | Measure | Description of analytic variable (metric/indicator) | Overall scoring | Number of articles for a given scoring method | Child age range[a] |
|---|---|---|---|---|---|
| Healthcare utilization | Vaccinations | Assessed whether child received specific vaccinations, total number of vaccinations received, or whether the child's vaccination was on schedule | Count, Binary | 5 | 0–59 months |
| Mortality | All-cause mortality | Maternal/caregiver reported and/or verified through verbal autopsy and/or based on administrative data. Neonatal, infant, or child death | Binary | 3 | 6–59 months |

[a] Child age range refers to the sample assessed across the studies using a given metric, and not necessarily the age range of children for whom the tool was developed or could be used.

### Health

We identified 102 total studies which reported at least one indicator for *health* across five categories: birth outcomes, morbidity, hygiene and health practices, healthcare utilization, and mortality (**Table 7**). Within the birth outcomes categories, although 4 of the indicators are based on international standards, these standards were not specifically reported or cited as the measurement source in the studies. Nevertheless, all birth outcomes indicators were consistently reported across multiple studies. Birth weight was the most frequently reported indicator (n = 41).

Eight out of the nine 9 of the morbidity indicators are based on international standards that were consistently reported across studies, with minor variability (e.g., change in recall period) to align the indicators for the purposes of the study. Child diarrhea was the most frequently reported indicator (n = 29).

With respect to hygiene and health practices, we identified two indicators. Neither one was standardized, and both assessed household- and child-level practices either combined or separately. Lastly, the healthcare utilization and mortality categories each contained only a single indicator. Neither one was standardized or consistently reported across studies.

### Discussion

This scoping review included 239 articles from over 50 LMICs that measured at least one outcome pertaining to nurturing care in a sample of caregivers and/or children younger than age five years. We identified several main measurement constructs for each nurturing care component. More specifically, this included: for early learning–stimulation practices, learning materials, and early childhood education; for responsive caregiving–measures specifically capturing responsive caregiving, quality of parent-child relationships more broadly, and responsive feeding; safety and security–disciplinary practices, maternal exposure to intimate partner violence, inadequate supervision, and birth registration; for nutrition–anthropometry, complementary feeding, breastfeeding, food security, and micronutrient status; and for health–birth outcomes, morbidity, hygiene and health practices, healthcare utilization, and mortality. Although the most common constructs were generally identifiable for each nurturing care component, we found greater variability in the definitions, measures, and specific indicators used for outcomes of early learning, responsive caregiving, and safety and security, compared to nutrition or health. Overall, this study provides a broad and comprehensive review of the current state of measurement of nurturing care and highlights the need for more research and guidance to inform robust standardized measures that are fit-for-purpose for monitoring and evaluating nurturing care globally.

There have been considerable efforts over the past decades to establish global recommendations and guidelines for child nutrition and health metrics and subsequent investment towards monitoring, accountability, and tracking of health and nutrition indicators for young children globally [12, 17]. We found that most of the nutrition and health indicators were multifunctional and used in both population-level household surveys and program evaluations. For example, minimum dietary diversity scores based on child consumption of any food in each of eight food groups has been broadly used across contexts, including as part of the DHS [11]. However, we found that one measure of child nutrition in particular– 24-hour dietary recall of types and quantities of all foods and beverages consumed–were used exclusively in program evaluations. This measure is more labor and time intensive, requires substantial training of enumerators, and may not often be feasible to collect as part of population-based household surveys [18].

In contrast, there was greater variation in definition, measures, and indicators used for outcomes of early learning, responsive caregiving, and safety and security. Of these three nurturing care components, the greatest heterogeneity and inconsistency was observed across measures of responsive caregiving. Most measures broadly assessed general qualities of parenting rather than specifically responsive caregiving [19, 20]. For example, the Observation of Mother-Child Interactions tool [21] was one of the most common standardized measures used for assessing parent-child interactions broadly. It comprises of 19 items (12 for parent behaviors and 7 for child behaviors). The original tool was developed to include three possible indicators: a parent-score, a child-score, or a total score, and the majority of all identified studies reported a total OMCI score or parent sub-score. Although six of the 12 parent items assess parental behaviors more relevant to responsiveness, none of the studies using the OMCI operationalized these as a specific indicator for responsive care. Thus, we did not classify the OMCI as a measure for specifically assessing responsive caregiving. Similar issues are present with the other measures that include observation of responsiveness alongside general parenting or parental engagement for early learning (e.g., HOME inventory). Given that subscales for responsiveness have not yet been established within broader measures of parenting, we found that the vast majority of current measures do not specifically assess responsiveness. This highlights the need for further measurement work, including the development and testing of a new tool, in order to fill this data gap in monitoring specifically responsive caregiving of the Nurturing Care Framework [22]. Notwithstanding, we found a stark increase in articles measuring responsive caregiving as well as early learning and safety and security over the past decade. These positive trends likely reflect the momentum and success of recent advocacy efforts and redoubling of investments in parenting programs for ECD in LMICs that have renewed interest and demand in measurement and evaluation of parenting outcomes with respect to ECD [23, 24].

At the same time, we also uncovered a number of methodological differences present across the landscape of nurturing care indicators. While most nutrition and health indicators (as well as many safety and security indicators) were scored as proportions (reflecting the primary intended design for use in population-level monitoring), outcome measures for early learning (e.g., FCI) and responsive caregiving (e.g., OMCI) were largely analyzed as continuous or index scores [21, 25]. Moreover, given the lack of validation studies for the optimal scoring of early learning and responsive caregiving measures, we identified inconsistencies with regards to the scoring, analytical approaches, and reporting of these indicators. For example, across studies measuring stimulation practices using the common measure of the FCI, we found substantial variation in methods and reporting of indicators, ranging from index scores, proportions that applied different cutoffs to the overall score, to individual indicators at the item-level. This heterogeneity in indicators of early learning and responsive care can also be seen

visually in the treemap, with each rectangle representing a unique scoring approach. Such analytic decisions and resulting indicators were largely not described or justified across studies.

Relatedly, evidence regarding reliability and validity was highly variable and not established for many scoring methods. Therefore, we could not directly compare the relative strength of the different indicators used across studies (e.g., using an index score versus cutoff to assess stimulation) or determine whether certain indicators demonstrated stronger reliability and validity across cultural contexts. Finally, while nearly all nurturing care indicators were used among children broadly under age 5 years, several of the measures were specifically developed or primarily assessed among children of more narrowly defined age ranges (e.g., HOME Inventory: Infant and Toddler version developed for children 0–3 years; OMCI developed for children aged 6–24 months). Therefore, additional measurement validation research is needed especially for measures of early learning and responsive caregiving measures to establish the psychometric properties and any adaptations need if to be used with a broad age range of children 0–5 years.

Taken together, our findings highlight the need for more research and guidance regarding the most valid and reliable measures, appropriate scoring methods, and standardized reporting of indicators for nurturing care. In particular, clear definitions–both in terms of the theoretical constructs as well as the analytic variable construction–are needed in order to operationalize and distinguish between the nurturing care components of early learning and responsive caregiving. Currently, due to the lack of established guidance for these nurturing care domains as well as suboptimal reporting of these measures in the peer-reviewed literature, we identified a considerable degree of uncertainty as to which measures adequately capture responsive caregiving. Standardized definitions and metrics are crucial for enabling robust monitoring and comparisons of nurturing care data across countries and time [1]. Recognizing the unique goals and varying constraints of population-level monitoring versus program evaluation or individual-level assessments, such measurement guidance and prioritized indicators for nurturing care should be tailored to these different purposes and contexts of measurement, as has similarly been proposed for measuring ECD for global monitoring versus program evaluation purposes in LMICs [26]. It is worth noting that nearly all identified measures of responsive caregiving were used in program evaluations or research cohort studies, largely involved direct observations, included multi-item scales, and generally required dedicated time for training and piloting with data collectors, which may not be as feasible to collect as part of large-scale surveys. More work is particularly needed to determine indicators that can be introduced into national surveys to ensure monitoring of responsive caregiving at a population-level. Lessons learned from the field of maternal and newborn health quality of care, and recent success with introducing direct observation indicators as part of large national surveys [27], can guide similar efforts towards potentially measuring responsive care in population-level surveys.

## Limitations

This scoping review had some limitations. First, articles varied in the reporting of measurement details (e.g., sources of measures, how indicators were constructed), which was partly a reflection of improved measurement reporting standards over time but also disciplinary differences in journal outlet expectations. As we relied on the information that was presented in the given article by the study authors, there is the possibility of misclassification for some measures and indicators of our results. At the same time, this approach was also advantageous in uncovering the heterogeneity in definitions and particular scoring approaches, especially for early learning and responsive caregiving. Second, we did not assess the feasibility considerations, cross-cultural applicability, or psychometric evidence (e.g., predictive validity with respect to

ECD outcomes) associated with the different measures and indicators, largely because these details were not reported in the majority of studies. These aspects are additionally critical for determining the relative strength and making decisions between different measurement approaches. Finally, our scoping review was limited to metrics reported in the peer-reviewed literature and in English language publications. While we may have missed other measures and indicators used in program reports or by implementing agencies, we expect that the measures and indicators summarized in our study include those that are the most robust and common approaches in the field.

## Conclusions

We reviewed the literature and identified measures and indicators used to assess outcomes relevant to the five domains of nurturing care for ECD. We uncovered significant variability with regards to measures, scoring, and reporting of indicators for particularly early learning, responsive caregiving, and safety and security. Based on our findings, there is a great need for further statistical analyses (e.g., validation, cross-cultural measurement invariance) as well as user-experience information (e.g., stakeholders' perceptions about relevance, feasibility, and practical considerations relating to administration) to guide subsequent processes of establishing the most optimal and robust indicators for use in LMIC contexts. While this current work has focused on measuring outcomes of nurturing care, monitoring guidance is needed to define and prioritize a standard set of input and output indicators that should also be comprehensively evaluated as part of the broader logic model for improving nurturing care.

## Supporting information

**S1 Checklist. PRISMA-ScR checklist.**
(DOCX)

**S1 Text. Search strategy.**
(DOCX)

**S1 Table. Characteristics for each study included in scoping review on nurturing care indicators.**
(DOCX)

## Author Contributions

**Conceptualization:** Joshua Jeong, Kathleen L. Strong, Bernadette Daelmans.

**Data curation:** Joshua Jeong, Lilia Bliznashka, Eileen Sullivan, Elizabeth Hentschel, Young-kwang Jeon.

**Formal analysis:** Joshua Jeong, Lilia Bliznashka, Eileen Sullivan, Elizabeth Hentschel, Young-kwang Jeon.

**Funding acquisition:** Joshua Jeong.

**Investigation:** Joshua Jeong.

**Methodology:** Joshua Jeong.

**Project administration:** Joshua Jeong.

**Supervision:** Joshua Jeong.

**Validation:** Joshua Jeong.

**Visualization:** Eileen Sullivan.

**Writing – original draft:** Joshua Jeong.

**Writing – review & editing:** Lilia Bliznashka, Eileen Sullivan, Elizabeth Hentschel, Young-kwang Jeon, Kathleen L. Strong, Bernadette Daelmans.

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
