## [Decision Letter · Decision Letter 0]

14 Feb 2022

PGPH-D-21-00200

Measurement tools and indicators for assessing nurturing care for early childhood development: a scoping review

Dear Dr. Jeong,

Thank you for submitting your manuscript to PLOS Global Public Health. After careful consideration, we feel that it has merit but does not fully meet PLOS Global Public Health’s publication criteria as it currently stands. Therefore, we invite you to submit a revised version of the manuscript that addresses the points raised during the review process.

We especially reccomend to:

1. revise the description of tools and scoring systems;

2. complete the description of the nurturing care;

3. consider including a flow diagram;

4. follow the detailed Reviewers comments presented below.

We look forward to receiving your revised manuscript.

Kind regards,

Hanna Nalecz, Ph.D.

Academic Editor

Journal Requirements:

1. Please provide separate figure files in .tif or .eps format only and ensure that all files are under our size limit of 20MB.

Additional Editor Comments (if provided):

Reviewers' comments:

Reviewer's Responses to Questions

**Comments to the Author**

1. Does this manuscript meet PLOS Global Public Health’s publication criteria? Is the manuscript technically sound, and do the data support the conclusions? The manuscript must describe methodologically and ethically rigorous research with conclusions that are appropriately drawn based on the data presented.

Reviewer #1: Yes

Reviewer #2: Yes

Reviewer #3: Yes

2. Has the statistical analysis been performed appropriately and rigorously?

Reviewer #1: Yes

Reviewer #2: Yes

Reviewer #3: Yes

3. Have the authors made all data underlying the findings in their manuscript fully available (please refer to the Data Availability Statement at the start of the manuscript PDF file)?

Reviewer #1: Yes

Reviewer #2: Yes

Reviewer #3: Yes

4. Is the manuscript presented in an intelligible fashion and written in standard English?

Reviewer #1: Yes

Reviewer #2: Yes

Reviewer #3: Yes

5. Review Comments to the Author

Reviewer #1: The authors have done very well in reviewing and delineating the pertinent literature in this scoping review of what is not just a very vast field but also a very important one. The manuscript itself is well written and organized and easy to understand and comprehend.

Manuscript pg 6; Methods section and search strategy: Please mention the time period during which the review was done

Table 4:

1) Time or duration of assessment are given for some observational tools but not for all of them. It would be advised that assessment time duration be included for all tools to ensure consistency.

2) Page 37: Please rephrase scoring description in Parent-child relationship (OMCI)

3) Page 44: Please rephrase scoring description for the entire item for reference number 53.

Please note that the above mentioned items on pages 37 and 44 have used the same wording as the source text. The authors are advised to revisit the statements and rephrase the description.

Reviewer #2: Re: Measurement tools and indicators for assessing nurturing care for early childhood

Development

Abstract

1. The authors state that there has been increasing political interest and evidence on effective programs …… It would be appropriate for the authors to contextualize their study to public health, not otherwise.

2. In the introduction, the authors mention the five distinct components of nurturing care. However, only four are mentioned. Please mention the fifth one too.

3. The first sentence of the results section fits to be in the methodology section of the abstract.

Main manuscript

Introduction

4. In the first line of paragraph 2 of the introduction, the authors specify the SDG(s) and corresponding targets(s) that is/are in line with the argument being put across.

5. In the 3rd paragraph, the authors mention about the five strategic action areas that are crucial for creating an enabling environment for families and caregivers to support young children’s development. Please state those strategic actions you are referring to.

6. It would be of help if you strengthen your introduction with some statistics around the magnitude of the problem.

7. In paragraph 4, you have not been specific enough about the whether the previous (scoping) reviews have only focused on health and nutrition, hence justifying yours which is rather broader.

Discussion

8. Line 352: the authors state that most measures assessed the general qualities of parenting rather than specifically responsive caregiving. It could be helpful if some sources justifying this statement are included her.

9. Line 376: Authors state that the outcome measures for early learning and responsive caregiving were largely analysed as continuous or index scores. Please cite the relevant sources supporting this statement.

10. Generally, in the discussion, only seven (7) sources were cited to back up the arguments. This is inadequate. There is need for authors to search for more relevant sources to back up their arguments.

Generally, the authors have done a fair job.

Reviewer #3: The diversity of methods and indicators for measuring nurturing care has been well described and demonstrated in this manuscript. The authors made an inventory of the tools and indicators used and showed the interest in standardisation. The topic seems relevant to me.

The manuscript can be accepted for publication in its current form.

However, the authors can clarify the criteria used to select 239 of the 3091 studies identified. Perhaps a diagram could better demonstrate this.

In lines 349-350, the authors mention greater variation in definition, measures, and indicators used for outcomes of early learning, responsive care giving, and safety and security. How has this been taken into account in this study, particularly in the discussion and limitations of the study?

6. PLOS authors have the option to publish the peer review history of their article (what does this mean?). If published, this will include your full peer review and any attached files.

**Do you want your identity to be public for this peer review?** For information about this choice, including consent withdrawal, please see our Privacy Policy.

Reviewer #1: No

Reviewer #2: **Yes: **David Kavuma

Reviewer #3: No

---

## [Decision Letter · Decision Letter 1]

31 Mar 2022

Measurement tools and indicators for assessing nurturing care for early childhood development: a scoping review

PGPH-D-21-00200R1

Dear Dr. Jeong,

We are pleased to inform you that your manuscript 'Measurement tools and indicators for assessing nurturing care for early childhood development: a scoping review' has been provisionally accepted for publication in PLOS Global Public Health.

Best regards,

Hanna Nalecz, Ph.D.

Academic Editor

Reviewer Comments (if any, and for reference):

Reviewer's Responses to Questions

**Comments to the Author**

1. If the authors have adequately addressed your comments raised in a previous round of review and you feel that this manuscript is now acceptable for publication, you may indicate that here to bypass the “Comments to the Author” section, enter your conflict of interest statement in the “Confidential to Editor” section, and submit your "Accept" recommendation.

Reviewer #1: All comments have been addressed

Reviewer #2: All comments have been addressed

2. Does this manuscript meet PLOS Global Public Health’s publication criteria? Is the manuscript technically sound, and do the data support the conclusions? The manuscript must describe methodologically and ethically rigorous research with conclusions that are appropriately drawn based on the data presented.

Reviewer #1: Yes

Reviewer #2: Yes

3. Has the statistical analysis been performed appropriately and rigorously?

Reviewer #1: Yes

Reviewer #2: Yes

4. Have the authors made all data underlying the findings in their manuscript fully available (please refer to the Data Availability Statement at the start of the manuscript PDF file)?

Reviewer #1: Yes

Reviewer #2: Yes

5. Is the manuscript presented in an intelligible fashion and written in standard English?

Reviewer #1: Yes

Reviewer #2: Yes

6. Review Comments to the Author

Reviewer #1: The authors have revised the article in line with the suggested changes.

Reviewer #2: The authors attended to the comments I raised. It would be better if the authors responded to my comments in a point-by-point response, instead of only working on the comments within the manuscript. Probably, the Editor would guide on this in the future.

7. PLOS authors have the option to publish the peer review history of their article (what does this mean?). If published, this will include your full peer review and any attached files.

**Do you want your identity to be public for this peer review?** For information about this choice, including consent withdrawal, please see our Privacy Policy.

Reviewer #1: No

Reviewer #2: No
